# PROTOTYPE-ASSISTED ADVERSARIAL LEARNING FOR UNSUPERVISED DOMAIN ADAPTATION

## ABSTRACT

This paper presents a generic framework to remedy the misalignment in unsupervised domain adaptation (UDA). Previous adversarial learning methods for UDA condition domain alignment only on pseudo labels, but pseudo labels may be inaccurate, hence bringing insufficient alleviation to the misalignment. Compared with pseudo labels, class prototypes are more accurate and reliable since they are summarized over all the instances and are able to represent the inherent semantic structures shared across domains. Therefore, we propose a novel Prototype-Assisted Adversarial Learning (PAAL) scheme, which incorporates instance probabilistic predictions and class prototypes together to provide reliable indicators for adversarial domain adaptation. With the PAAL scheme, we align both the instance feature representations and class prototypes to alleviate the misalignment among semantically different instances. Also, we exploit the class prototypes as proxy to minimize the intra-class variance in the target domain to mitigate the misalignment among semantically similar instances. With these novelties, we constitute a Prototype-Assisted Conditional Domain Adaptation (PACDA) framework which achieves state-of-the-art results on two UDA tasks, i.e., object recognition (*Office-Home*, *ImageCLEF-DA*, and *Office*) and synthetic-to-real semantic segmentation (*GTA5→Cityscapes* and *Synthia→Cityscapes*).

## 1 INTRODUCTION

Unsupervised domain adaptation (UDA) aims to leverage the knowledge of a labeled data set (source domain) to help train a predictive model for a unlabeled data set (target domain). Deep UDA methods bring noticeable performance gain to many tasks (Long et al., 2015; Saito et al., 2017; Richter et al., 2016; Tsai et al., 2018; Lee et al., 2019; Vu et al., 2019a) by exploiting supervision from heterogeneous sources. Some methods exploit maximum mean discrepancy (MMD) (Gretton et al., 2008; Long et al., 2015) or other distribution statistics like central moments (Sun & Saenko, 2016; Zellinger et al., 2017; Koniusz et al., 2017) for domain adaptation. Recently, generative adversarial learning (Goodfellow et al., 2014) provides a promising alternative solution to UDA problem.

Since the labels of the target instances are not given in UDA, adversarial learning scheme for adaptation (Ganin & Lempitsky, 2015) suffers from the cross-domain misalignment, where the target instances from a class A are potentially misaligned with source instances from another class B. Inspired by the pseudo-labeling strategy from semi-supervised learning, previous methods either used the pseudo labels in the target domain to perform joint distribution discrepancy minimization (Long et al., 2013; 2015) or developed conditional adversarial learning methods that involve one high-dimensional domain discriminator (Long et al., 2018) or multiple discriminators (Chen et al., 2017b; Pei et al., 2018). Though effective, these conditional domain adversarial learning methods align different instances from different domains relying only on their own predictions. Simple probabilistic predictions or pseudo labels may not accurately represent the semantic information of input instances, misleading the alignment. A toy example is given in Fig. 1(a). The pseudo label of the chosen instance $x$ is inclined to be class 'square' while the ground truth label is class 'circle'. Only guided by the instance prediction, the 'circle' class in the target domain and the 'square' class in the source domain are easily confused, causing the misalignment in the adversarial domain adaptation.

To remedy the misalignment, we propose to exploit the class prototypes for adversarial domain alignment, instead of using only the possibly inaccurate predictions. Prototypes are global feature representations of different classes and are relevant to the inherent semantic structures shared across

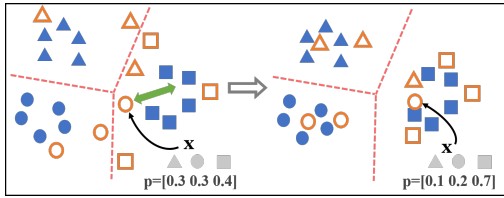
(a) conditional adversarial learning

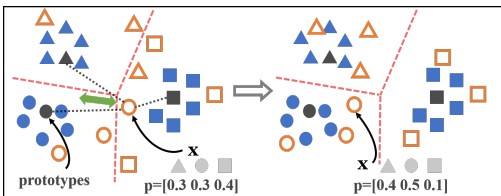
(b) prototype-assisted adversarial learning

Figure 1: Illustration of two adversarial learning schemes. Different from class-agnostic adversarial learning that pursues the marginal distribution alignment but ignores the semantic consistency, (a) conditional adversarial learning relies heavily on the instance-level pseudo labels to perform conditional distribution alignment, while (b) our prototype-assisted adversarial learning integrates the instance-level pseudo labels and global class prototypes to make the conditional indicators more reliable. Class information is denoted in different shapes with source in solid and target in hollow.

domains. As shown in Fig. 1(b), class prototypes are expected to remedy the negative effects of inaccurate probabilistic predictions. Motivated by this, we propose a Prototype-Assisted Adversarial Learning (PAAL) scheme which complements instance predictions with class prototypes to obtain more reliable conditional information for guiding the source-target feature representation alignment.

Specifically, we summarize the class prototypes from all instances according to their predictions. In this way, on one hand, we lower the dependence of class prototypes on instance predictions which may be inaccurate, and on the other hand, we encourage the instances with greater certainty to contribute more to their corresponding class prototypes. The prototypes are updated dynamically through a moving average strategy to make them more accurate and reliable. Then by broadcasting class prototypes to each instance according to its probability prediction, the inaccurate semantic distribution depicted by instance predictions can be alleviated. Based on reliable prototype-based conditional information, we align both the instance feature representations and the class prototypes through the proposed PAAL scheme to relieve the alignment among semantically dissimilar instances. However, such a conditional domain alignment may promote the confusion among semantically similar instances across domains to some degree. To further alleviate it, we introduce an intra-class objective in the target domain to pursue the class compactness. Built on the proposed PAAL scheme and this intra-class compactness objective, we develop a Prototype-Assisted Conditional Domain Adaptation (PACDA) framework for solving UDA problems. Extensive experimental evaluations on both object recognition and semantic segmentation tasks clearly demonstrate the advantages of our approaches over previous state-of-the-arts (Long et al., 2018; Xu et al., 2019; Luo et al., 2019; Tsai et al., 2019).

The contributions of this work can be summarized into three folds: 1) To the best of our knowledge, we are the first to leverage the class prototypes in conditional adversarial learning to prevent the misalignment in UDA; 2) We propose a simple yet effective domain adversarial learning framework PACDA to remedy the misalignment among semantically similar instances as well as semantically dissimilar instances; 3) The proposed PAAL scheme and PACDA framework are generic, and our framework achieves the state-of-the-art results on several unsupervised domain adaptation tasks including object recognition and semantic segmentation.

## 2 RELATED WORK

**Unsupervised Domain Adaptation.** UDA is first modeled as the covariate shift problem (Shimodaira, 2000) where marginal distributions of different domains are different but their conditional distributions are the same. To address it, (Dudík et al., 2006; Huang et al., 2007) exploit a nonparametric instance re-weighting scheme. Another prevailing paradigm (Pan et al., 2010; Long et al., 2013; Herath et al., 2017) aims to learn feature transformation with some popular cross-domain metrics, e.g., the empirical maximum mean discrepancy (MMD) statistics. Recently, a large number of deep UDA works (Long et al., 2015; Haeusser et al., 2017; Saito et al., 2018; Tsai et al., 2018) have been developed and boosted the performance of various vision tasks. Generally, they can be divided into discrepancy-based and adversarial-based methods. Discrepancy-based methods (Tzeng et al., 2014; Long et al., 2017) address the dataset shift by mitigating specific discrepancies defined on different layers of a shared model between domains, e.g. resembling shallow feature transforma-

tion by matching higher moment statistics of features from different domains (Zellinger et al., 2017; Koniusz et al., 2017). Recently, adversarial learning has become a dominantly popular solution to domain adaptation problems. It leverages an extra domain discriminator to promote domain confusion. (Ganin & Lempitsky, 2015) designs a gradient reversal layer inside the classification network and (Tzeng et al., 2017) utilizes an inverted label GAN loss to fool the discriminator.

**Pseudo-labeling.** UDA can be regarded as a semi-supervised learning (SSL) task where unlabeled data are replaced by the target instances. Therefore, some popular SSL strategies, e.g., entropy minimization (Grandvalet & Bengio, 2005; Vu et al., 2019b), mean-teacher (Tarvainen & Valpola, 2017; French et al., 2018), and virtual adversarial training (Miyato et al., 2018; Shu et al., 2018), have been successfully applied to UDA. Pseudo-labeling is favored by most UDA methods due to its convenience. For example, (Saito et al., 2017; Li et al., 2019) exploit the intermediate pseudo-labels with tri-training and self-training, respectively. (Pan et al., 2019) obtains target-specific prototypes with the help of pseudo labels and aligns prototypes across domains at different levels. Recently, curriculum learning (Choi et al., 2019), self-paced learning (Zou et al., 2018) and re-weighting schemes (Long et al., 2018) are further leveraged to tackle possible false pseudo-labels.

**Conditional Domain Adaptation.** Apart from the explicit integration with the last classifier layer, pseudo-labels can also be incorporated into adversarial learning to enhance the feature-level domain alignment. Concerning shallow methods (Long et al., 2013; Zhang et al., 2017), pseudo-labels can help mitigate the joint distribution discrepancy via minimizing multiple class-wise MMD measures. (Long et al., 2017) proposes to align the joint distributions of multiple domain-specific layers across domains based on a joint maximum mean discrepancy criterion. Recently, (Chen et al., 2017b; Pei et al., 2018) leverages the probabilities with multiple domain discriminators to enable fine-grained alignment of different data distributions in an end-to-end manner. In contrast, (Long et al., 2018) conditions the adversarial domain adaptation on discriminative information via the outer product of feature representation and classifier prediction. Motivated by the semantically-consistent GAN, (Cicek & Soatto, 2019) imposes a multi-way adversarial loss instead of a binary one on the domain alignment. However, these methods all highly rely on the localized pseudo-labels to align label-conditional feature distributions and ignore the global class-level semantics. As far as we know, we are the first to exploit class prototypes to guide the domain adversarial learning. Compared with (Pei et al., 2018; Long et al., 2018), our PACDA framework complements the original feature representations with reliable semantic features and merely involves two low-dimensional domain discriminators, making the domain alignment process simple, conditional, and reliable.

## 3 METHOD

In this section, we first begin with the basic settings of UDA and then give detailed descriptions on the proposed PAAL scheme and the PACDA framework. Though proposed for image classification, they can also be easily applied to semantic segmentation.

### 3.1 PROBLEM SETTINGS

In a vanilla UDA task, we are given label-rich source domain data $\{(\boldsymbol{x}_s^i, \boldsymbol{y}_s^i)\}_{i=1}^{n_s}$ sampled from the joint distribution $P_s(\boldsymbol{x}_s, \boldsymbol{y}_s)$ and unlabeled target domain data $\{\boldsymbol{x}_t^i\}_{i=1}^{n_t}$ sampled from the joint distribution $Q_t(\boldsymbol{x}_t, \boldsymbol{y}_t)$, where $\boldsymbol{x}_s^i \in \mathcal{X}_S$ and $\boldsymbol{y}_s^i \in \mathcal{Y}_S$ denote an image and its corresponding label from the source domain dataset, $\boldsymbol{x}_t^i \in \mathcal{X}_T$ denotes an image from the target domain dataset and $P_s \neq Q_t$. The goal of UDA is to learn a discriminative model from $\mathcal{X}_S$, $\mathcal{Y}_S$, and $\mathcal{X}_T$ to predict labels for unlabeled target samples $\mathcal{X}_T$.

As described in (Ganin et al., 2016), a vanilla domain adversarial learning framework consists of a feature extractor network $G$, a classifier network $F$, and a discriminator network $D$. Given an image $\boldsymbol{x}$, we denote the feature representation vector extracted by $G$ as $\boldsymbol{f} = G(\boldsymbol{x}) \in \mathbb{R}^d$ and the probability prediction obtained by $F$ as $\boldsymbol{p} = F(\boldsymbol{f}) \in \mathbb{R}^c$ where $d$ means the feature dimension and $c$ means the number of classes. The vanilla domain adversarial learning method in (Ganin et al., 2016) can be formulated as optimizing the following minimax optimization problem:

$$\min_{G,F} \max_D \mathcal{L}_y(G, F) - \lambda_{adv}\mathcal{L}_{adv}(G, D), \tag{1}$$

$$\mathcal{L}_{adv}(G, D) = -\mathbb{E}_{\boldsymbol{x}_s^i \sim P_s} \log[D(\boldsymbol{f}_s^i)] - \mathbb{E}_{\boldsymbol{x}_t^j \sim Q_t} \log[1 - D(\boldsymbol{f}_t^j)], \tag{2}$$

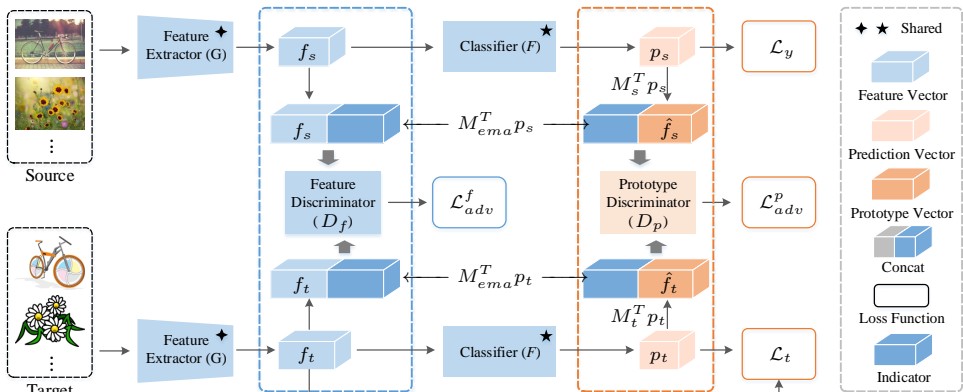

Figure 2: Overview of the proposed PACDA framework which consists of a shared feature extractor $G$, a shared classifier $F$, and two domain discriminators ($D_f$, $D_p$). $M_{ema}$ represents the global class prototype matrix while $M_{s,t}$ is computed by source or target instances within current batch.

$$\mathcal{L}_y(G, F) = -\mathbb{E}_{(\boldsymbol{x}_s^i, \boldsymbol{y}_s^i) \sim P_s} \, \boldsymbol{y}_s^{i\,T} \log(\boldsymbol{p}_s^i), \quad \boldsymbol{p}_s^i = F(G(\boldsymbol{x}_s^i)), \tag{3}$$

where the binary domain classifier $D : \mathbb{R}^d \to [0, 1]$ predicts the domain assignment probability over the input features, $\mathcal{L}_y(G, F)$ is the cross-entropy loss of source domain data as for the classification task, and $\lambda_{adv}$ is the trade-off parameter.

## 3.2 PROTOTYPE-ASSISTED ADVERSARIAL LEARNING (PAAL) SCHEME

The misalignment in UDA of multi-class distributions challenges the popular vanilla adversarial learning. In previous works (Long et al., 2017; Pei et al., 2018; Long et al., 2018), target domain data are conditioned only on corresponding pseudo labels predicted by the model for adversarial domain alignment. The general optimization process of these methods is the same as aforementioned vanilla domain adversarial learning, except that feature representations jointly with predictions are considered by the discriminator $D$:

$$\mathcal{L}_{adv}^{cond}(G, D) = -\mathbb{E}_{\boldsymbol{x}_s^i \sim P_s} \log[D(\boldsymbol{f}_s^i, \boldsymbol{p}_s^i)] - \mathbb{E}_{\boldsymbol{x}_t^j \sim Q_t} \log[1 - D(\boldsymbol{f}_t^j, \boldsymbol{p}_t^j)], \tag{4}$$

is the conditional adversarial loss that leverages the classification predictions $\boldsymbol{p}_s$ and $\boldsymbol{p}_t$. A classic previous work (Long et al., 2018) implicitly conditions the feature representation on the prediction through the outer product $\boldsymbol{f} \otimes \boldsymbol{p}$, and uses one shared discriminator to align the conditioned feature representations. (Long et al., 2018) further proves that using the outer product can perform much better than simple concatenation $\boldsymbol{f} \oplus \boldsymbol{p}$. Different from (Long et al., 2018), (Chen et al., 2017b; Pei et al., 2018) explicitly utilize multiple class-wise domain discriminators to align the feature representations relying on the corresponding predictions.

However, the pseudo labels may be inaccurate due to the domain shift. Therefore, only conditioning the alignment on pseudo labels can not safely remedy the misalignment. Compared with the pseudo labels, the class prototypes are more robust and reliable in terms of representing the shared semantic structures (Yang et al., 2018). To acquire more reliable and accurate conditional information for domain adversarial learning, we propose to complement instance predictions with class prototypes and reformulate the adversarial loss to:

$$\mathcal{L}_{adv}^{paal}(G, D) = -\mathbb{E}_{\boldsymbol{x}_s^i \sim P_s} \log[D(\boldsymbol{f}_s^i, \boldsymbol{M}_{ema}^T \, \boldsymbol{p}_s^i)] - \mathbb{E}_{\boldsymbol{x}_t^j \sim Q_t} \log[1 - D(\boldsymbol{f}_t^j, \boldsymbol{M}_{ema}^T \, \boldsymbol{p}_t^j)]. \tag{5}$$

Here $\boldsymbol{M}_{ema} \in \mathbb{R}^{c \times d}$ denotes the global class prototype matrix in our prototype-assisted adversarial learning loss $\mathcal{L}_{adv}^{paal}(G, D)$. In reality, the reliable conditional information is obtained through broadcasting the global class prototypes to each independent instance according to its prediction $\boldsymbol{p}$.

We propose to summarize feature representations of the instances within the same class as the corresponding prototype. Then the probability prediction is leveraged to obtain accurate class prototypes. Using predictions as weights can adaptively control the contributions of typical and non-typical instances to the class prototype, making class prototypes more reliable. Specifically, we first gather the feature representation of each instance relying on its prediction to generate the batch-level class prototypes. Then the global class prototypes can be obtained by virtue of an averaging strategy such

as exponential moving average (ema) on the batch ones. This process can be formulated as

$$\boldsymbol{M}_{ema} = \lambda_{ema}\boldsymbol{M}_{ema} + (1 - \lambda_{ema})\boldsymbol{M}, \text{where } \boldsymbol{M} = [\boldsymbol{m}_1, \cdots, \boldsymbol{m}_c], \ \boldsymbol{m}_k = \sum_i^n \boldsymbol{p}^{k,i} \boldsymbol{f}^{i^T} / \sum_i^n \boldsymbol{p}^{k,i}. \quad (6)$$

Here $n$ means the batch size, $\boldsymbol{p}^{k,i}$ represents the probability of the $i$-th instance belonging to the $k$-th semantic class, $\lambda_{ema}$ is an empirical weight, $\boldsymbol{M} \in \mathbb{R}^{c \times d}$ is the batch-level class prototype matrix and $\boldsymbol{M}_{ema}$ is the global one computed by certain source domain data and contributes to more reliable conditional information exploited by discriminators. Similarly, batch-level class prototypes are broadcast to each instance in this batch through $\boldsymbol{M}_a^T \boldsymbol{p}_a$ which can be denoted as $\widehat{\boldsymbol{f}}_a, a \in \{s, t\}$.

## 3.3 Prototype-Assisted Conditional Domain Adaptation (PACDA) Framework

With our prototype-based conditional information, we further propose a Prototype-Assisted Conditional Domain Adaptation (PACDA) framework. This framework aligns both instance-level and prototype-level feature representations through PAAL and promotes the intra-class compactness in target domain such that the misalignment can be substantially alleviated even though no supervision is available in the target domain. Its overall architecture is shown in Fig. 2.

Besides the backbone feature extractor $G$ and the task classifier $F$, there are two discriminators in our framework PACDA, i.e., the instance-level feature discriminator $D_f$ and the prototype-level feature discriminator $D_p$. We can formulate our general objective function as (w.l.o.g., $\mathcal{L}_{adv} \leftarrow \mathcal{L}_{adv}^{paal}$),

$$\mathcal{L}(\mathcal{X}_S, \mathcal{Y}_S, \mathcal{X}_T) = \mathcal{L}_y - \lambda_{adv}^f \mathcal{L}_{adv}^f - \lambda_{adv}^p \mathcal{L}_{adv}^p + \lambda_t \mathcal{L}_t, \quad (7)$$

where $\lambda$ denotes balance factors among different loss functions, $\mathcal{L}_y$ is the supervised classification loss on source domain data described by Eq. (3), $\mathcal{L}_{adv}^f$ is the adversarial loss to align instance feature representations across domains, $\mathcal{L}_{adv}^p$ is the adversarial loss to align class prototype representations across domains, and $\mathcal{L}_t$ is the loss to promote the intra-class compactness in target domain.

**Instance-Level Alignment** Conditioning the instance feature representation on our prototype-based conditional information, we seek to align feature representations across domains at the instance-level through discriminator $D_f$. With the assistance of the accurate semantic structures embedded in class prototypes, misalignment among semantically dissimilar instances can be effectively alleviated. We can define the instance-level adversarial loss $\mathcal{L}_{adv}{}^f$ as

$$\mathcal{L}_{adv}^f(G, F, D_f) = -\mathbb{E}_{\boldsymbol{x}_s^i \sim P_s} \log[D_f(\boldsymbol{f}_s^i \oplus \boldsymbol{M}_{ema}^T \boldsymbol{p}_s^i)] - \mathbb{E}_{\boldsymbol{x}_t^j \sim Q_t} \log[1 - D_f(\boldsymbol{f}_t^j \oplus \boldsymbol{M}_{ema}^T \boldsymbol{p}_t^j)]. \quad (8)$$

**Prototype-Level Alignment** Instance-level alignment only implicitly aligns the multi-class distribution across domains, which may not ensure the semantic consistency between two domains. Besides, since in practice global class prototypes are collected from only source domain data, which possibly cannot accurately represent inherent semantic structures in the target domain due to the domain shift. Taking into account these two causes, we perform the prototype-level alignment with discriminator $D_p$ to explicitly align the class prototype representations across domains. The specific loss function $\mathcal{L}_{adv}^f$ is defined as

$$\mathcal{L}_{adv}^p(G, F, D_p) = -\mathbb{E}_{\boldsymbol{x}_s^i \sim P_s} \log[D_p(\widehat{\boldsymbol{f}}_s^i \oplus \boldsymbol{M}_{ema}^T \boldsymbol{p}_s^i)] - \mathbb{E}_{\boldsymbol{x}_t^j \sim Q_t} \log[1 - D_p(\widehat{\boldsymbol{f}}_t^j \oplus \boldsymbol{M}_{ema}^T \boldsymbol{p}_t^j)]. \quad (9)$$

**Intra-Class Compactness** Although adversarial alignment based on PAAL can relieve the misalignment among obviously semantically different instances, it cannot well handle the misalignment among semantically similar instances. Specifically, incorporating class prototypes into instance predictions would confuse semantically similar instances during domain alignment and result in the misalignment among them. To solve this problem, our framework further promotes the intra-class compactness in the target domain to enlarge the margin between instances of semantically similar classes. Taking the prototypes as proxy, we minimize the following loss for target domain samples to encourage the intra-class compactness:

$$\mathcal{L}_t(G, F) = \mathbb{E}_{\boldsymbol{x}_t^j \sim Q_t} \|\boldsymbol{f}_t^j - \widehat{\boldsymbol{f}}_t^j\|_2^2. \quad (10)$$

Thus, the complete minimax optimization problem of our PACDA framework can be formulated as

$$\min_{G,F} \max_{D_f, D_p} \mathcal{L}_y(G, F) - \lambda_{adv}^f \mathcal{L}_{adv}^f(G, F, D_f) - \lambda_{adv}^p \mathcal{L}_{adv}^p(G, F, D_p) + \lambda_t \mathcal{L}_t(G, F). \quad (11)$$

With only two low-dimensional ($2 \times d$) discriminators added, we effectively remedy the misalignment in domain adversarial learning. Some theoretical insights with the help of domain adaptation theory (Ben-David et al., 2010) is discussed in the Appendix.

## 4 EXPERIMENTS

### 4.1 EXPERIMENTAL SETUP

We conduct experiments to verify the effectiveness and generalization ability of our methods, i.e., PACDA (full) in Eq. (11) and PAAL ($\lambda_{adv}^p = \lambda_t = 0$) on two different UDA tasks, including cross-domain object recognition on *ImageCLEF-DA*[1], *Office31* (Saenko et al., 2010) and *Office-Home* (Venkateswara et al., 2017), and synthetic-to-real semantic segmentation for *GTA5* (Richter et al., 2016)→*Cityscapes* (Cordts et al., 2016) and *Synthia* (Ros et al., 2016)→ *Cityscapes*.

**Datasets.** *Office-Home* is a new challenging dataset that consists of 65 different object categories found typically in 4 different Office and Home settings, i.e., Artistic (**Ar**) images, Clip Art (**Ca**), Product images (**Pr**), and Real-World (**Re**) images. *ImageCLEF-DA* is a standard dataset built for the 'ImageCLEF2014:domain-adaptation' competition. We follow (Long et al., 2015) to select 3 subsets, i.e., **C**, **I**, and **P**, which share 12 common classes. *Office31* is a popular dataset that includes 31 object categories taken from 3 domains, i.e., Amazon (**A**), DSLR (**D**), and Webcam (**W**).

*Cityscapes* is a realistic dataset of pixel-level annotated urban street scenes. We use its original training split and validation split as the training target data and testing target data respectively. *GTA5* consists of 24,966 densely labeled synthetic road scenes annotated with the same 19 classes as *Cityscapes*. For *Synthia*, we take the SYNTHIA-RAND-CITYSCAPES set as the source domain, which is composed of 9,400 synthetic images compatible with annotated classes of *Cityscapes*.

**Implementation Details.** For object recognition, we follow the standard protocol (Ganin & Lempitsky, 2015), i.e. using all the labeled source instances and all the unlabeled target instances for UDA, and report the average accuracy based on three random trials for fair comparisons. Following (Long et al., 2018; Xu et al., 2019), we experiment with ResNet-50 model pretrained on ImageNet. Specifically, we follow (Long et al., 2018) to choose the network parameters, and all convolutional layers and the classifier layer are trained through backpropagation, where $\lambda_t$=5e-3, $\lambda_{ema}$=5e-1, $\lambda_{adv}^f$ and $\lambda_{adv}^p$ increase from 0 to 1 with the same strategy as (Ganin & Lempitsky, 2015). Regarding the domain discriminator, we design a simple two-layer classifier (256→1024→1) for both $D_f$ and $D_p$. Empirically, we fix the batch size to 36 with the initial learning rate being 1e-4.

For semantic segmentation, we adopt DeepLab-V2 (Chen et al., 2017a) based on ResNet-101 (He et al., 2016) as done in (Tsai et al., 2018; Vu et al., 2019b; Luo et al., 2019; Tsai et al., 2019). Following DCGAN (Radford et al., 2015), the discriminator network consists of three $4 \times 4$ convolutional layers with stride 2 and channel numbers $\{256, 512, 1\}$. In training, we use SGD (Bottou, 2010) to optimize the network with momentum (0.9), weight decay (5e-4), and initial learning rate (2.5e-4). We use the same learning rate policy as in (Chen et al., 2017a). Discriminators are optimized by Adam (Kingma & Ba, 2015) with momentum ($\beta_1 = 0.9$, $\beta_2 = 0.99$), initial learning rate (1e-4) along with the same decreasing strategy as above. For both tasks, $\lambda_{adv}^f$ is set to 1e-3 following (Tsai et al., 2018) and $\lambda_{ema}$ is set to 0.7. For *GTA5*→*Cityscapes*, $\lambda_{adv}^p$=1e-3 and $\lambda_t$=1e-5. For *Synthia*→*Cityscapes*, $\lambda_{adv}^p$=1e-4 and $\lambda_t$=1e-4.

All experiments are implemented via **PyTorch** on a single Titan X GPU. The total iteration number is set as 10k for object recognition and 100k for semantic segmentation. For objection recognition tasks, we choose the hyper-parameters which have the minimal mean entropy of target data (Morerio et al., 2018) on Ar→Cl for convenience. For semantic segmentation tasks, training split of *Cityscapes* is used for the hyper-parameters selection. Data augmentation skills like random scale or random flip and ten-crop ensemble evaluation are not adopted.

### 4.2 COMPARISON RESULTS

**Cross-Domain Object Recognition.** The comparison results between our methods (i.e., PAAL and PACDA) and state-of-the-art (SOTA) approaches (Xu et al., 2019; Long et al., 2018; Zhang et al., 2018) on *Office-Home*, *Office31*, and *ImageCLEF-DA* are shown in Tables 1 and 2, respectively. As indicated in these tables, PACDA improves previous approaches in the average accuracy for all three benchmarks (e.g., 67.3%→68.7% for *Office-Home*, 88.1%→88.8% for *ImageCLEF-DA*, and 87.7%→89.3% for *Office31*). Generally, PACDA performs the best for most transfer tasks. Taking

---
[1] https://www.imageclef.org/2014/adaptation

Table 1: Accuracy (%) on *Office-Home* for UDA under ResNet-50. **Red**: Best, *Blue*: Second best.

| Methods | Ar→Cl | Ar→Pr | Ar→Re | Cl→Ar | Cl→Pr | Cl→Re | Pr→Ar | Pr→Cl | Pr→Re | Re→Ar | Re→Cl | Re→Pr | Avg. |
|---|---|---|---|---|---|---|---|---|---|---|---|---|---|
| ResNet-50 (He et al., 2016) | 34.9 | 50.0 | 58.0 | 37.4 | 41.9 | 46.2 | 38.5 | 31.2 | 60.4 | 53.9 | 41.2 | 59.9 | 46.1 |
| DANN (Ganin & Lempitsky, 2015) | 45.6 | 59.3 | 70.1 | 47.0 | 58.5 | 60.9 | 46.1 | 43.7 | 68.5 | 63.2 | 51.8 | 76.8 | 57.6 |
| CDAN (Long et al., 2018) | 49.0 | 69.3 | 74.5 | 54.4 | 66.0 | 68.4 | 55.6 | 48.3 | 75.9 | 68.4 | 55.4 | 80.5 | 63.8 |
| CDAN+E (Long et al., 2018) | 50.7 | 70.6 | 76.0 | 57.6 | 70.0 | 70.0 | 57.4 | 50.9 | 77.3 | 70.9 | *56.7* | 81.6 | 65.8 |
| DWT-MEC (Roy et al., 2019) | 50.3 | 72.1 | 77.0 | 59.6 | 69.3 | 70.2 | 58.3 | 48.1 | 77.3 | 69.3 | 53.6 | *82.0* | 65.6 |
| SAFN (Xu et al., 2019) | *52.0* | *73.3* | *77.9* | **65.2** | *71.5* | *73.2* | *63.6* | *52.6* | *78.2* | *72.3* | **57.1** | 81.5 | *67.3* |
| PAAL | 50.7 ±0.2 | 69.2 ±0.5 | 73.2 ±0.2 | 58.2 ±0.5 | 66.4 ±0.6 | 68.3 ±0.5 | 55.5 ±0.3 | 48.3 ±0.4 | 74.5 ±0.1 | 68.5 ±0.2 | 55.6 ±0.3 | 79.2 ±0.2 | 64.0 |
| PACDA | **53.5** ±0.2 | **73.5** ±0.1 | **78.4** ±0.2 | *64.1* ±0.4 | **73.2** ±0.3 | **74.4** ±0.2 | **64.1** ±0.5 | *50.8* ±0.9 | **79.9** ±0.1 | **73.6** ±0.1 | 56.6 ±0.1 | **82.6** ±0.2 | **68.7** |

Table 2: Accuracy (%) on *ImageCLEF-DA* and *Office31* for UDA under ResNet-50.

| Datasets / Methods | *ImageCLEF-DA* | | | | | | Avg. | *Office31* | | | | | | Avg. |
|---|---|---|---|---|---|---|---|---|---|---|---|---|---|---|
| | C→I | C→P | I→C | I→P | P→C | P→I | | A→D | A→W | D→A | D→W | W→A | W→D | |
| ResNet-50 (He et al., 2016) | 78.0 | 65.5 | 91.5 | 74.8 | 91.2 | 83.9 | 80.7 | 68.9 | 68.4 | 62.5 | 96.7 | 60.7 | 99.3 | 76.1 |
| DANN (Ganin & Lempitsky, 2015) | 87.0 | 74.3 | 96.2 | 75.0 | 91.5 | 86.0 | 85.0 | 79.7 | 82.0 | 68.2 | 96.9 | 67.4 | 99.1 | 82.2 |
| CDAN (Long et al., 2018) | 90.5 | 74.5 | 97.0 | 76.7 | 93.5 | 90.6 | 87.1 | 89.8 | 93.1 | 70.1 | 98.2 | 68.0 | **100.** | 86.6 |
| CDAN+E (Long et al., 2018) | 91.3 | 74.2 | **97.7** | 77.7 | 94.3 | 90.7 | 87.7 | 92.9 | 94.1 | 71.0 | *98.6* | 69.3 | **100.** | *87.7* |
| iCAN (Zhang et al., 2018) | 89.9 | **78.5** | 94.7 | **79.5** | 92.0 | 89.7 | 87.4 | 90.1 | 92.5 | 72.1 | **98.8** | 69.9 | **100.** | 87.2 |
| CAT (Deng et al., 2019) | 91.3 | 75.3 | 95.5 | 77.2 | 93.6 | 91.0 | 87.3 | 90.8 | 94.4 | *72.2* | 98.0 | *70.2* | **100.** | 87.6 |
| SAFN (Xu et al., 2019) | 91.1 | *77.0* | 96.2 | 78.0 | 94.7 | *91.7* | *88.1* | 87.7 | 88.8 | 69.8 | 98.4 | 69.7 | 99.8 | 85.7 |
| PAAL | *91.4* ±0.1 | 74.6 ±0.6 | *97.4* ±0.1 | 77.6 ±0.4 | *95.2* ±0.1 | 90.7 ±0.4 | 87.8 | *93.4* ±0.0 | *94.6* ±0.7 | 70.2 ±0.8 | 97.7 ±0.1 | 69.9 ±0.6 | **100.** ±0.0 | 87.6 |
| PACDA | **92.2** ±0.2 | 76.2 ±0.2 | **97.7** ±0.1 | *78.1* ±0.2 | **96.3** ±0.2 | **92.4** ±0.2 | **88.8** | **94.8** ±0.0 | **95.7** ±0.3 | **72.8** ±1.5 | 98.0 ±0.2 | **74.4** ±0.5 | *99.8* ±0.0 | **89.3** |

Table 3: Comparison results of synthetic-to-real semantic segmentation using the same architecture with NonAdapt and AdaptSeg (Tsai et al., 2018), AdvEnt (Vu et al., 2019b), CLAN (Luo et al., 2019) and AdaptPatch (Tsai et al., 2019). **Top**: *GTA5 → Cityscapes*. **Bottom**: *Synthia → Cityscapes*.

| | Methods | road | sdwk | bldng | wall | fence | pole | light | sign | veg. | ter. | sky | per. | rider | car | truck | bus | train | mbike | bike | mIoU |
|---|---|---|---|---|---|---|---|---|---|---|---|---|---|---|---|---|---|---|---|---|---|
| source: *GTA5* | NonAdapt | 75.8 | 16.8 | 77.2 | 12.5 | 21.0 | 25.5 | 30.1 | 20.1 | 81.3 | 24.6 | 70.3 | 53.8 | 26.4 | 49.9 | 17.2 | 25.9 | 6.5 | 25.3 | **36.0** | 36.6 |
| | AdaptSeg | 86.5 | 25.9 | 79.8 | 22.1 | 20.0 | 23.6 | 33.1 | 21.8 | 81.8 | 25.9 | 75.9 | 57.3 | 26.2 | 76.3 | 29.8 | 32.1 | **7.2** | *29.5* | *32.5* | 41.4 |
| | AdvEnt | *89.9* | 36.5 | 81.6 | 29.2 | **25.2** | 28.5 | 32.3 | 22.4 | *83.9* | 34.0 | 77.1 | 57.4 | 27.9 | 83.7 | 29.4 | 39.1 | 1.5 | 28.4 | 23.3 | 43.8 |
| | CLAN | 87.0 | 27.1 | 79.6 | 27.3 | 23.3 | 28.3 | 35.5 | **24.2** | 83.6 | 27.4 | 74.2 | 58.6 | *28.0* | 76.2 | *33.1* | 36.7 | *6.7* | **31.9** | 31.4 | 43.2 |
| | AdaptPatch | 89.2 | *38.4* | 80.4 | 24.4 | 21.0 | 27.7 | 32.9 | 16.1 | 83.1 | 34.1 | *77.8* | 57.4 | 27.6 | 78.6 | 31.2 | 40.2 | 4.7 | 27.6 | 27.6 | 43.2 |
| | PAAL | 89.7 | 30.3 | *81.7* | *30.7* | *24.8* | *29.4* | **36.3** | 22.6 | 83.4 | *36.1* | 76.3 | **60.4** | *28.4* | *83.8* | 32.7 | *41.7* | 4.4 | 27.3 | 31.0 | *44.8* |
| | PACDA | **92.4** | **50.3** | **83.3** | **33.2** | 24.6 | **32.9** | *36.1* | *22.7* | **84.4** | **38.0** | *79.9* | *60.3* | 24.7 | **85.2** | **37.8** | **46.6** | **7.2** | 26.4 | 18.6 | **46.6** |
| source: *Synthia* | NonAdapt | 55.6 | 23.8 | 74.6 | - | - | - | 6.1 | *12.1* | 74.8 | - | 79.0 | 55.3 | 19.1 | 39.6 | - | 23.3 | - | 13.7 | 25.0 | 38.6 |
| | AdaptSeg | 79.2 | 37.2 | 78.8 | - | - | - | *9.9* | 10.5 | 78.2 | - | 80.5 | 53.5 | 19.6 | 67.0 | - | 29.5 | - | *21.6* | 31.3 | 45.9 |
| | AdvEnt | **87.0** | **44.1** | 79.7 | - | - | - | 4.8 | 7.2 | **80.1** | - | **83.6** | **56.4** | **23.7** | 72.7 | - | 32.6 | - | 12.8 | *33.7* | 47.6 |
| | CLAN | 81.3 | 37.0 | *80.1* | - | - | - | **16.1** | **13.7** | 78.2 | - | 81.5 | 53.4 | *21.2* | 73.0 | - | 32.9 | - | **22.6** | 30.7 | 47.8 |
| | AdaptPatch | 82.2 | 39.4 | 79.4 | - | - | - | 6.5 | 10.8 | 77.8 | - | 82.0 | 54.9 | 21.1 | 67.7 | - | 30.7 | - | 17.8 | 32.2 | 46.3 |
| | PAAL | **87.2** | *43.2* | 80.0 | - | - | - | 8.2 | 9.6 | 79.2 | - | *82.3* | *56.1* | 20.3 | *81.1* | - | *33.5* | - | 16.5 | 32.3 | *48.4* |
| | PACDA | *87.0* | 42.4 | **80.8** | - | - | - | 9.7 | 10.9 | *80.0* | - | **83.6** | 50.1 | 20.7 | **82.1** | - | **34.4** | - | 19.4 | **38.6** | **49.2** |

a careful look at PAAL, we find that it always beats CDAN and achieves competitive performance with SOTA methods like CAT (Deng et al., 2019).

**Synthetic-to-real Semantic Segmentation.** We compare PAAL and PACDA with SOTA methods (Tsai et al., 2018; Vu et al., 2019b; Luo et al., 2019; Tsai et al., 2019) on synthetic-to-real semantic segmentation. Following (Chen et al., 2017b), we evaluate models on all 19 classes for *GTA5→Cityscapes* while on only 13 classes for *Synthia→Cityscapes*. As shown in Table 3, without bells and whistles, our PAAL method outperforms all of those methods and our PACDA framework further achieves new SOTA results on both tasks, i.e., 43.8%→46.6% for *GTA5→Cityscapes* and 47.8%→49.2% for *Synthia→Cityscapes* in terms of the mean IoU (mIoU) value.

### 4.3 FURTHER ANALYSIS

**Quantitative Analysis.** To verify the effectiveness of each component in Eq. (11), we introduce a variant named $PAAL_{f,p}$ that merely ignores the intra-class objective ($\lambda_t = 0$). The empirical convergence curves about Ar→Cl in Fig. (3)(a) imply that all of our variants tend to converge after 10k iterations, and the second term can help accelerate the convergence. Fig. 3(b) shows that all terms

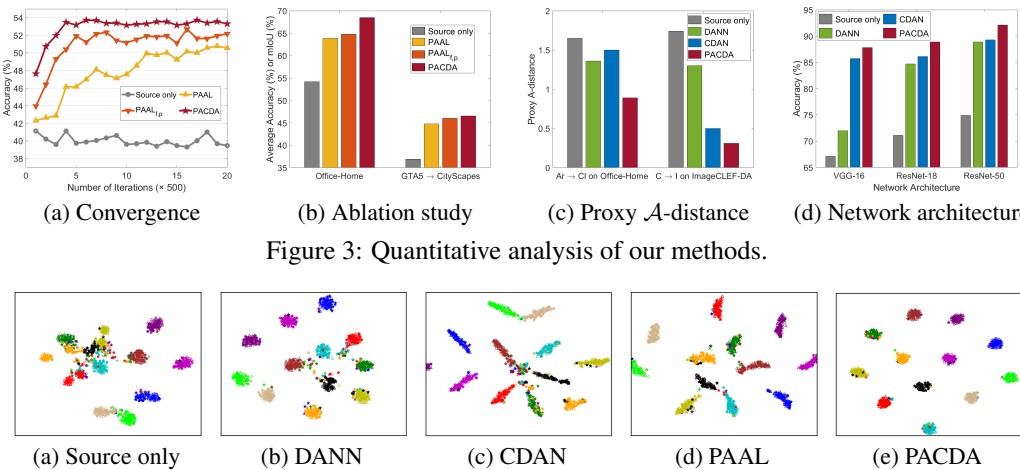

(a) Convergence     (b) Ablation study     (c) Proxy $\mathcal{A}$-distance     (d) Network architecture

Figure 3: Quantitative analysis of our methods.

(a) Source only     (b) DANN     (c) CDAN     (d) PAAL     (e) PACDA

Figure 4: t-SNE (Maaten & Hinton, 2008) embedding visualization of UDA methods for C→I on *ImageCLEF-DA* (class information is denoted by different colors with source in △ with target in ⋆).

in the PACDA framework, i.e., PAAL alignment at different levels and the intra-class objective, can bring evident improvement on both tasks. As shown in Fig. (3)(c), we provide the proxy $\mathcal{A}$-distances (Ganin et al., 2016) of different methods for Ar→Cl and C→I. The $\mathcal{A}$-distance $Dist_{\mathcal{A}}=2(1-2\epsilon)$ is a popular measure for domain discrepancy, where $\epsilon$ is the test error of a binary classifier trained on the learned features. All the UDA methods have smaller distances than 'source only' by aligning different domains. Besides, our PACDA has the minimum distance for both tasks, implying that it can learn better features to bridge the domain gap between domains. To testify the sensitivity of our PACDA, in Fig. (3)(d) we report the accuracies of DANN, CDAN and PACDA for C→I on the ImageCLEF-DA with 3 different backbone architectures, i.e., VGG-16, ResNet-18, and ResNet-50. Obviously, PACDA is the best-performing method that shows desirable robustness when the network changing from VGG-16 to ResNet-18.

**Qualitative Analysis.** For object recognition, we study the t-SNE visualizations of aligned features generated by different UDA methods in Fig. 4. As expected, conditional methods including CDAN and PAAL can semantically align multi-class distributions much better than DANN. Besides, PAAL learns slightly better features than CDAN due to less misalignment. Once considering the intra-class objective, PACDA further enhances PAAL by pushing away semantically confusing classes, which achieves the best adaptation performance. For semantic segmentation, we present some qualitative results in Fig. 5. Similarly, PAAL effectively improves the adaptation performance and PAAL$_{f,p}$ as well as PACDA can further improve the segmentation results.

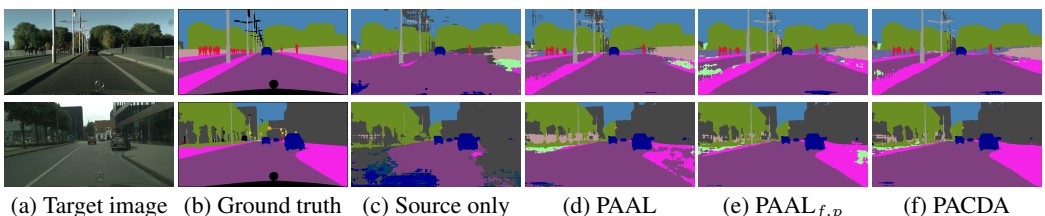

(a) Target image    (b) Ground truth    (c) Source only    (d) PAAL    (e) PAAL$_{f,p}$    (f) PACDA

Figure 5: Qualitative results of synthetic-to-real semantic segmentation for *GTA5→Cityscapes*.

## 5 CONCLUSION

In this work, we developed the prototype-assisted adversarial learning scheme to remedy the misalignment for UDA tasks. Unlike previous conditional ones whose performance is vulnerable to inaccurate instance predictions, our proposed scheme leverages the reliable and accurate class prototypes for aligning multi-class distributions across domains and is demonstrated to be more effective to prevent the misalignment. Then we further augment this scheme by imposing the intra-class compactness with the prototypes as proxy. Extensive evaluations on both object recognition and semantic segmentation tasks clearly justify the effectiveness and superiority of our UDA methods over well-established baselines.

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

# A APPENDIX

## A.1 THEORETICAL INSIGHTS

We try to explain why our PAAL works well for UDA according to the domain adaptation theory proposed in (Ben-David et al., 2010). Denote by $\epsilon_P(F) = \mathbb{E}_{(f,y)\in P}[F(f) \neq y]$ the risk of a classifier model $F \in \mathcal{H}$ w.r.t. the distribution $P$, and by $\epsilon_P(F_1, F_2) = \mathbb{E}_{(f,y)\in P}[F_1(f) \neq F_2(f)]$ the disagreement between hypotheses $F_1, F_2 \in \mathcal{H}$. Particularly, (Ben-David et al., 2010) gives a well-known upper bound on the target risk $\epsilon_Q(F)$ of classifier $F$ in the following,

$$\epsilon_Q(F) \leq \epsilon_P(F) + [\epsilon_P(F^*) + \epsilon_Q(F^*)] + |\epsilon_P(F, F^*) - \epsilon_Q(F, F^*)|, \tag{12}$$

where $F^*$ is the ideal classifier induced from $F^* = \arg\min_{F\in\mathcal{H}}[\epsilon_P(F) + \epsilon_Q(F)]$, and the last term is related to the classical $\mathcal{H}$-divergence $d_{\mathcal{H}\Delta\mathcal{H}}(P,Q) = 2\sup_{F,F^*\in\mathcal{H}} |\epsilon_P(F, F^*) - \epsilon_Q(F, F^*)|$. Besides, according to (Ben-David et al., 2010), the empirical $\mathcal{H}$-divergence calculated by $m$ respective samples from distributions $P$ and $Q$ converges uniformly to the true $\mathcal{H}$-divergence for classifier classes $\mathcal{H}$ of finite VC dimension $d$, which is expressed as

$$d_{\mathcal{H}\Delta\mathcal{H}}(P,Q) \leq \hat{d}_{\mathcal{H}\Delta\mathcal{H}}(P,Q) + 4\sqrt{\frac{d\log(2m) + \log(2/\delta)}{m}}. \tag{13}$$

The work (Ganin & Lempitsky, 2015) introduces a binary domain discriminator to minimize the empirical $\mathcal{H}$-divergence $\hat{d}_{\mathcal{H}\Delta\mathcal{H}}(P,Q)$, which aligns the marginal distributions well. However, if two multi-class distributions $P$ and $Q$ are not semantically aligned, there may not be any classifier with low risk in both domains, which means the second term of the upper bound in Eq. (12) is very large. The proposed PAAL scheme leverages reliable conditional information in the adversarial learning module so that semantically similar samples from different domains are implicitly aligned, thus it has a high possibility of decreasing the second term. Compared with (Long et al., 2018), the input to domain the adversarial learning module is much more compact ($2 \times d \ll c \times d$), which helps decrease the second term in Eq. (13).

