# OpenReview forum: "PROTOTYPE-ASSISTED ADVERSARIAL LEARNING FOR UNSUPERVISED DOMAIN ADAPTATION"
_ICLR.cc/2020/Conference — Reject_

### Official Review · AnonReviewer1 · 2019-10-07
**Official Blind Review #1**

**Rating:** 3

**Review:**

This paper proposes to leverage prototypes to solve the mismatch problem in unsupervised domain adaptation. It further imposes intra-class compactness to help ambiguous classes. Experiments show it achieves new state-of-the-art results in several datasets.

pros:
+ intra-class compactness to help ambiguous classes

concerns:
-- Prototypes does not come from nowhere. They come from predictions. If you worry about the quality of target predictions (pseudo labels), then Eq. 8 and Eq. 9 are questionable. The intra-class compactness relies on p_t, too. The authors should explain why prototypes are superior than pseudo labels in [1].
-- How does the authors select hyper-parameters? There are lots of magic numbers in Section 4.1 about hyper-parameters but no clues about how to tune them. Recently there is a paper [2] about model selection for UDA, maybe the authors should try it.

details:
- terminology: "intra-class" is better than "within class"
- separate citations: e.g. entropy minimization, mean-teacher, and virtual adversarial training, have been successfully applied to UDA (Vu et al., 2019; French et al., 2018; Shu et al., 2018) -> entropy minimization (Vu et al., 2019), mean-teacher (French et al., 2018), and virtual adversarial training (Shu et al., 2018), have been successfully applied to UDA
- confusion: At the last of Section 3.2, it says \hat{f}=M^{T}p. But in Eq. 9, \hat{f} and M^{T}p are concatenated, which is confusing: why do you concatenate two identical vectors?
- Implementation Details: Section 4.1, paragraph 4: \lambda^{f}_{adv} =5e-3, \lambda^{f}_{adv} and \lambda^{p}_{adv} increase from 0 to 1. It is confusing that \lambda^{f}_{adv} both is a constant and changes continuously.

[1] Conditional adversarial domain adaptation, Long et.al, in NeurIPS 2018
[2] Towards Accurate Model Selection in Deep Unsupervised Domain Adaptation, You et.al, in ICML 2019

**Experience Assessment:**

I have published in this field for several years.

**Review Assessment: Checking Correctness Of Derivations And Theory:**

I did not assess the derivations or theory.

**Review Assessment: Checking Correctness Of Experiments:**

I carefully checked the experiments.

**Review Assessment: Thoroughness In Paper Reading:**

I read the paper thoroughly.

---

> ### Author Response · Authors · 2019-11-12
> **Reply1 to AnonReviewer1**
>
> We thank the reviewer for useful comments and constructive advice. We would like to make the following clarifications.
>
>
> 1. Why are prototypes superior than pseudo labels in [1]?
>
> The label prediction of single target instance may be inaccurate due to the domain shift. Thus, only conditioning the adversarial alignment on pseudo labels would make the results easy to be misled by inaccurate predictions. In [1], the authors propose CDAN to condition the domain adversarial alignment on predictions through outer product, which is superior than simply concatenating features and predictions.
>
> Prototypes are summarized only from source instances and dynamically updated by moving average strategy along with the feature learning process. Thus, prototypes are more accurate and reliable to represent the shared semantic structures. Therefore, taking prototypes to assist the domain alignment would be more robust to inaccurate instance-wise predictions.
>
> We conducted following experiments to support this. We reproduce related methods in [1] including CDAN, CDAN+E, DANN-[f,g], and compared them with variants of our PAAL method including PAAL, PAAL+E. "+E", i.e., the entropy conditioning in [1]. Both DANN-[f,g] and CDAN only condition the domain adversarial alignment on predictions. Our PAAL method only replaces the original predictions g in DANN-[f,g] by \M_{ema}^{T}g. For a comprehensive comparison, we add the entropy conditioning in [1] to get PAAL+E and then compare PAAL+E with CDAN+E.
>
> All experiments use the codes released by [1] and share the same hyperparameters provided by [1]. Below we report the average results (mean accuracy%(std)) of different methods on Office-Home based on three random trials without ten-crop ensemble evaluation.
>
> ---------------------------------------------------------------------------------------------------------------------------------------------------------------
> methods |   DANN-[f,g] [1]|  |      CDAN       [1]  |      CDAN+E     [1]  |      PAAL    (ours)  |      PAAL+E  (ours)
> ---------------------------------------------------------------------------------------------------------------------------------------------------------------
> Ar->Cl               44.1(0.4)                   51.3(0.2)                 51.7(0.2)                    50.7(0.2)                      53.0(0.2)
> Ar->Pr               58.5(0.5)                  67.3(0.3)                  68.9(0.3)                    69.2(0.5)                      70.6(0.5)
> Ar->Re              67.5(0.2)                   73.6(0.2)                 74.7(0.2)                    73.2(0.2)                      75.1(0.2)
> Cl->Ar               47.6(0.6)                   54.8(0.5)                 57.0(0.4)                     58.2(0.5)                      59.6(0.7)
> Cl->Pr               57.5(0.3)                   65.0(0.5)                 68.2(0.3)                     66.4(0.6)                      69.4(1.0)
> Cl->Re              59.9(0.4)                   68.0(0.2)                 69.7(0.2)                     68.3(0.5)                      69.7(0.2)
> Pr->Ar              47.3(0.3)                   54.4(0.5)                 57.4(0.5)                     55.5(0.3)                       57.6(0.5)
> Pr->Cl              40.6(0.7)                    46.4(0.5)                 49.5(0.3)                     48.3(0.4)                      50.7(0.4)
> Pr->Re             70.0(0.5)                    73.9(0.3)                 75.7(0.1)                     74.5(0.1)                      76.5(0.2)
> Re->Ar             61.4(0.1)                    66.5(0.3)                 68.8(0.1)                     68.5(0.2)                      70.7(0.1)
> Re->Cl              49.9(0.2)                    53.1(0.3)                55.5(0.2)                      55.6(0.3)                      57.0(0.3)
> Re->Pr             76.0(0.4)                    78.9(0.3)                 80.2(0.2)                     79.2(0.2)                       81.9(0.2)
> ---------------------------------------------------------------------------------------------------------------------------------------------------------------
> Avg.                    56.7                            62.8                          64.8                            64.0                              66.0
> ---------------------------------------------------------------------------------------------------------------------------------------------------------------
>
> Evidently, benefiting from the reliable prototypes, PAAL performs much better than DANN-[f,g]. PAAL also consistently outperforms CDAN for both with and without entropy reweighting (+E).

---

> ### Author Response · Authors · 2019-11-12
> **Reply2 to AnonReviewer1**
>
> 2. How does the author select hyper-parameters in Section 4.1?
>
> We observed values of different losses during the training process, then tried some empirical values of hyper-parameters. For object recognition tasks, we chose the hyper-parameters which have the minimal entropy on target data, following [3]. For semantic segmentation, we used the training split of Cityscapes for UDA training and validation split for testing. We directly chose the hyper-parameters which perform the best on the training split of Cityscapes. Indeed we found our model performance is robust to the hyper parameter choice.  We have added the explanations into revision.
>
> Taking the \lambda_t and \lambda_{ema} in object recognition tasks as an example, we explain how we selected hyper-parameters. We tried \lambda_ema with values including 0.3, 0.5 and 0.7. For \lambda_t, we tried 1e-4, 5e-4, 1e-3, 5e-3, 1e-2 and 5e-2. We compared the results (mean entropy(accuracy%)) at the last iteration on the first task of Office-Home. i.e., Ar to Cl.
>
>
> ---------------------------------------------------------------------------------------------------------------------------------------------------------------
> \lambda_{ema} \\ \lambda_t    |    1e-4         |    5e-4         |    1e-3        |    5e-3         |    1e-2         |     5e-2
> ------------------------------------------------------------------------------------- --------------------------------------------------------------------------
> 0.3                                                  | 0.60(50.6)  | 0.56(49.7)  | 0.51(51.8)  | 0.47(52.4)  | 0.60(48.9)  |  4.04(5.4)
> 0.5                                                  | 0.51(51.9)  | 0.52(52.0)  | 0.50(53.1)  | 0.45(53.2)  | 0.61(49.2)  |  2.75(33.1)
> 0.7                                                  | 0.47(52.7)  | 0.50(51.9)  | 0.48(52.8)  | 0.47(53.6)  | 0.51(51.0)  |  4.0(4.9)
> ---------------------------------------------------------------------------------------------------------------------------------------------------------------
>
> Because \lamba_t=5e-3 and \lambda_{ema}=0.5 has the lowest mean entropy, we finally adopted this hyperparameter setting for all object recognition tasks.
>
> We also think the model selection is especially important for UDA and [2] provides an effective method to accurately select models. We are happy to try this method in the future.
>
> 3.  About the details:
>
> -terminology: We agree and have replaced "within-class" by "intra-class".
>
> -separate citations: We agree and have separately cited the original methods as well as their applications in UDA tasks.
>
> -confusion: As for your confusion about Eq. (9). \hat{f}=M^{T}p at the last sentence of Section 3.2 broadcasts the batch-level class prototypes to each instance within current batch. While \M_{ema}^{T}p broadcasts the global class prototypes to each instance for the conditional domain adversarial adaptation. \M_{ema}^{T}p is always used as the reliable conditional information for alignment, which complements instance predictions with global prototypes. By formulating Eq. (9), we aim to explicitly align the batch-level class prototypes \hat{f}=M^{t}p across domains, considering reasons stated in Section 3.3 Prototype-Level Alignment.
>
> -Implementation Details: Thanks for pointing out the typos. We have corrected them. In section 4.1, we mistook the notations for the weight \lambda_t of the intra-class objective in object recognition and semantic segmentation. In paragraph 4: \lamda^{f}_{adv} = 5e-3 should be \lambda_t = 5e-3. In paragraph 5, for GTA2Cityscapes, \lambda^{t}_{adv} = 1e-5 should be \lambda_t = 1e-5. Another typo error in paragraph 5, for Synthia2Cityscapes, \lambda^{t}_{adv} = 1e-4 should be \lambda_t = 1e-4.
>
> [1] Conditional adversarial domain adaptation, Long et.al, in NeurIPS 2018
> [2] Towards Accurate Model Selection in Deep Unsupervised Domain Adaptation, You et.al, in ICML 2019
> [3] Minimal-Entropy Correlation Alignment for Unsupervised Deep Domain Adaptation, Morerio et.al, in ICLR 2018

---

### Official Review · AnonReviewer2 · 2019-10-23
**Official Blind Review #2**

**Rating:** 3

**Review:**

Summary:
- key problem: address "class mismatch" in adversarial learning methods for unsupervised domain adaptation (UDA);
- contributions: 1) extension of the domain adversarial learning objective to leverage class prototypes (exponential moving average of features weighted by predicted class probabilities) in addition to pseudo-labels and intermediate representations (cf. eqs.5-11), 2) state-of-the-art results on several UDA tasks (Office-Home, ImageCLEF-DA, sim2real on Cityscapes).

Recommendation: weak accept (with some reservations below).

Key reason: interesting and effective use of prototypes for UDA.
- The formulation of the prototypes and additional learning objectives for UDA are clear and seem novel, although I would like to see a discussion of additional related works:
-- "Mean teachers are better role models: Weight-averaged consistency targets improve semi-supervised deep learning results", Tarvainen and Valpola, NeurIPS'17;
-- "Unsupervised Domain Adaptation with Similarity Learning", Pinheiro, CVPR'18;
-- "Transferable Prototypical Networks for Unsupervised Domain Adaptation", Pan et al, CVPR'19.
- The effectiveness of the contributions is validated on multiple UDA tasks, and the ablative analysis supports the claims (that prototype-level alignment and within-class compactness helps).

Main reservation: the specific problem is not clearly formalized.
- What is the often mentioned but not clearly described "class mismatch" problem in UDA? To the best of my knowledge, this not a standard problem (could not find any mention in the previous literature, no citations or definitions in the submission). Is it that the target label space is different than the source label space (e.g., different ontologies)? In this case, what is the information on the target label space that enables unsupervised adaptation from the source one? What is the inductive bias / prior / assumptions?
- Alternatively, is the tackled problem only the noise in the pseudo-labels?
- In any case, the submission would greatly benefit from a clearer mathematical formalism and experimental characterization of the specific problem tackled here, especially in light of claims like "conditioning the alignment on pseudo labels can not well address the mismatch problem. Compared with the pseudo labels, the class prototypes are more robust and reliable in terms of representing the distribution of different semantic classes."

Additional Feedback:
- missing references on sim2real UDA: "DADA: Depth-aware Domain Adaptation in Semantic Segmentation" (Vu et al, ICCV'19), "SPIGAN: Privileged Adversarial Learning from Simulation" (Lee et al, ICLR'19)

## Post rebuttal update

I would like to thank the authors for replying to our questions. The clarifications with respect to related works and missing references is helpful, although a bit high-level (i.e. not necessarily describing the relative advantages of the proposed method). Nonetheless, the expected benefits of prototypes is still not entirely clear enough here, for instance regarding the main statistical assumptions that the method needs to make to get robust prototypes (e.g., in the presence of outliers or specific forms of "inaccuracies" in the pseudo-labels or "domain misalignment"). Therefore, due to the overall lack of mathematical clarity in the text and rebuttal, my main reservation remains, and I will change my "weak accept / borderline" score to weak reject. I encourage the authors to formalize the problem in a clearer, non-ambiguous way, discussing more explicitly the limitations of the proposed method.

**Experience Assessment:**

I have published in this field for several years.

**Review Assessment: Checking Correctness Of Derivations And Theory:**

I assessed the sensibility of the derivations and theory.

**Review Assessment: Checking Correctness Of Experiments:**

I assessed the sensibility of the experiments.

**Review Assessment: Thoroughness In Paper Reading:**

I read the paper at least twice and used my best judgement in assessing the paper.

---

> ### Author Response · Authors · 2019-11-12
> **Reply1 to AnonReviewer2**
>
> We thank the reviewer for the insightful comments. As for your review, we have the following explanations.
>
> 1. Discuss additional related works.
>
> Thanks for your reminder. We have added them into revision. Here we discuss the mentioned works [1,2,3] as below.
>
> “Mean Teacher” in [1] proposes to average weights of consecutive students models to get a more accurate model. In this way, the quality of teacher-generated targets would be improved. The similarity between “Mean Teacher” and our work is that we both use the exponential moving average (EMA). “Mean Teacher” uses EMA for consistency regularization, while we use EMA because the limited batch size cannot involve enough instances to obtain accurate class prototypes.
>
> “SimNet” in [2] replaces the fully-connected classifier in [4] with a similarity-based classifier. Moreover, prototypes are learned separately with the feature learning backbone.
>
> “TPN” in [3] extends the framework of “prototypical networks” in [5] for UDA, but obtains three types of class prototypes, i.e., source-specific prototypes, target-specific prototypes and shared prototypes. "TPN" tries to reduce the domain discrepancy both at class-level and sample-level. Class-level alignment independently pushes prototypes of the same class to be close. Sample-level alignment simultaneously aligns all prototypes by enforcing that score distributions by different classifiers (prototypes) for each sample should be consistent. “TPN” is similar to our work because we both use prototypes for UDA and enforce the domain alignment at class-level (prototype-level) and sample-level (instance-level). The main difference is that we utilize source prototypes only for assisting domain adversarial training and [3] attempts to learn transferable prototypes for inference by matching prototypes at different levels.
>
> 2. What is the often mentioned but not clearly described "class mismatch" problem in UDA? (No citations or definitions in the submission.)
>
> The “class mismatch” problem we mentioned means that the unlabeled target instances are misclassified in UDA with multi-class distribution due to that the unlabeled target instance from class A may be misaligned with source instance from class B in terms of their representations. This especially challenges the global domain adversarial alignment in [4], where although global statistics across domains may be aligned, instances of different classes may still be misaligned [6,7,8].
>
> 3. Is it that the target label space is different than the source label space?
>
> Sorry for the confusion. Our work is proposed to address UDA where the label space is assumed to be shared across domains.
>
> 4. Is the tackled problem only the noise in the pseudo-labels?
>
> No, the tackled problem is the misalignment in UDA with multi-class distribution, especially in  the adversarial domain alignment. Conditional domain alignment is an effective method to tackle this problem, where class-wise alignment is achieved by conditioning the adversarial learning on predictions [6,7].  Pseudo labels may be inaccurate and would mislead the domain adversarial alignment. Fortunately, reliable class prototypes can be obtained to represent the multi-class distribution shared across domains. As a result, we propose to utilize prototypes to assist the domain adversarial alignment. We do not explicitly or only address the noise within pseudo labels. We hope prototypes-assisted adversarial learning can mitigate misalignment among semantically dissimilar instances. And with prototypes as proxy, the intra-class objective can further reduce the misalignment among semantically similar instances.
>
> We add the entropy-aware weight of instance predictions in [6] during adversarial alignment to our PAAL method noted as PAAL+E, which suppresses effects by the noise within pseudo-labels. Experiments in "Reply1 to AnonReviewer1" show that PAAL+E brings another 2% improvement of accuracy, which means PAAL can be improved further by explicitly suppressing the noise within pseudo-labels. Experiments also demonstrate that compared with CDAN/CDAN+E [6] which only condition on pseudo labels, our PAAL methods bring consistent improvement.

---

> ### Author Response · Authors · 2019-11-12
> **Reply2 to AnonReviewer2**
>
> 5. The claims like "conditioning...semantic classes" not clear.
>
> We agree with you and understand that your main concern is about the unclear expression of the specific tackled problem. We have carefully revised the content about the tackled problem to make it clearer.
>
> 6. Additional feedback: missing references.
>
> We have added the missing references, i.e., [9] and [10], to the discussion on applications of UDA tasks.
>
>
> [1] "Mean teachers are better role models: Weight-averaged consistency targets improve semi-supervised deep learning results", Tarvainen and Valpola, NeurIPS'17;
> [2] "Unsupervised Domain Adaptation with Similarity Learning", Pinheiro, CVPR'18;
> [3] "Transferable Prototypical Networks for Unsupervised Domain Adaptation", Pan et al, CVPR'19.
> [4] “Unsupervised Domain Adaptation by Backpropagation” Ganin and Lempitsky, ICML’15;
> [5] “Prototypical Networks for Few-shot Learning”, Snell et al, NeurIPS'17.
> [6] "Conditional adversarial domain adaptation", Long et al, NeurIPS'18
> [7] "No More Discrimination: Cross City Adaptation of Road Scene Segmenters", Chen et al, ICCV'17
> [8] "FCNs in the Wild: Pixel-level Adversarial and Constraint-based Adaptation", Hoffman et al, arXiv:1612.02649
> [9] "DADA: Depth-aware Domain Adaptation in Semantic Segmentation", Vu et al, ICCV'19
> [10] "SPIGAN: Privileged Adversarial Learning from Simulation", Lee et al, ICLR'19

---

### Author Response · Authors · 2019-11-13
**General Note**


We thank both reviewers for their useful comments, which help us to refine our paper. Thanks to these comments, we make the following changes in our updated submission.

1. We have modified the expression of the specific tackled problem from “class mismatch” to “misalignment” that is widely used in previous UDA works.

2. We have explained how we select hyper-parameters for our UDA methods in Section 4.1.

3. We have modified the expression of “noisy and inaccurate pseudo labels” to “inaccurate pseudo labels” to make it clearer that our work mainly aims to remedy effects by misleading information conveyed by inaccurate pseudo labels rather than the noise within them.

4. We have modified the expression “within-class” to “intra-class” to make it clearer.

5. We have corrected some typos about hyper-parameters pointed out by Review#1 in Section 4.1.

6. We have added related works mentioned by Review#2 in Section 1 and Section 2 .

7. We have used separate citations pointed out by Review#1 in Section 2.

We hope the reviewers will be satisfied with these revisions and detailed responses below.

---

### Decision · Program_Chairs · 2019-12-19

**Decision:**

Reject

**Comment:**

The paper focuses on adversarial domain adaptation, and proposes an approach inspired from the DANN. The contribution lies in additional terms in the loss, aimed to i) align the source and target prototypes  in each class (using pseudo labels for target examples); ii) minimize the variance of the latent representations for each class in the target domain.

Reviews point out that the expected benefits of target prototypes might be ruined if the pseudo-labels are too noisy; they note that the specific problem needs be more clearly formalized and they regret the lack of clarity of the text. The sensitivity w.r.t. the hyper-parameter values needs be assessed more thoroughly.

One also notes that SAFN is one of the baseline methods; but its best variant (with entropic regularization) is not considered, while the performance thereof is on par or greater than that of PACFA for ImageCLEF-Da; idem for AdapSeg (consider its multi-level variant) or AdvEnt with MinEnt.

For these reasons, the paper seems premature for publication at ICLR 2020.